# Interpretable Time-Series and Trajectory Analysis for Detection of Autism-Related Motor Behaviors

**Olga Isaeva**
Russian Academy of Science
Krasnoyarsk, Russia
isaeva@icm.krasn.ru

**Ivan Maltsev, Anna Varlamova**
Russian Academy of Science
Moscow, Russia
{ivan.malz, varlamowa.anuta2015}@yandex.ru

**Sergey Strizhak**
Russian Academy of Science
Moscow, Russia
strizhak@yandex.ru

**Konstantin Gnidko, Olga Krivorotko**
Sirius University
Sirius Federal Territory, Russia
{gnidko.ko, krivorotko.oi}@talantiuspeh.ru

## Abstract

Early detection of Autism Spectrum Disorder (ASD) is crucial, as intervention effectiveness strongly depends on the age at diagnosis. ASD is characterized by stereotypical motor behaviors, many of which involve cyclic patterns such as hand flapping, circular walking, or whole-body rocking. We analyze 103,000 frames extracted from 66 expert-recorded videos of 36 children in controlled playroom settings. Child pose is estimated using YOLOv11Pose, and behavioral localization is performed via sliding-window segmentation of skeletal keypoint trajectories. Given the cyclic structure of stereotypies, trajectory analysis is reduced to interpretable kinematic and spectral descriptors capturing circularity, repetition, and periodicity. Circular motion is described through radius statistics and curvature, repetition through PCA-based segment similarity, and periodicity through FFT- and autocorrelation-derived features such as dominant frequency and energy ratios. These features are used to train classical machine learning models and a multilayer perceptron. Performance is evaluated using grouped 5-fold cross-validation and reported in terms of Accuracy, Recall, and F1-score. For hand flapping detection, gradient boosting achieved the best performance (Accuracy 0.947, F1 0.950), while for whole-body rocking it achieved Accuracy 0.918 and F1 0.872. Feature contributions are analyzed using SHAP to ensure model interpretability.

## 1 Introduction

In recent years, large-scale video models and multimodal architectures based on transformers and foundation models have become increasingly prominent in the automatic analysis of behaviors in children with Autism Spectrum Disorder (ASD). Deng et al. (2025) proposed a multimodal audio-visual framework for ASD behavior recognition that integrates visual, acoustic, and speech features using pretrained models such as CLIP (Radford et al., 2021), ImageBind (Chen et al., 2023) and Whisper (Radford et al., 2023). Their AV-ASD dataset incorporates publicly available resources, including the Self-Stimulatory Behavior Dataset (SSBD) (Rajagopalan et al., 2013), which contains video clips of predefined stereotypical behaviors such as flapping, rocking, and spinning. The dataset consists of 569 videos with a total duration of approximately 6 hours and 40 minutes. This work demonstrates strong classification performance and reflects the growing shift toward large-scale multimodal learning. However, such end-to-end architectures typically require considerable computational resources and substantial training data.

A related direction is explored by Hu et al. (2024), where a Video Swin Transformer (Liu et al., 2022) is augmented with textual descriptions of behavioral categories. Textual supervision is incorporated

as an auxiliary signal to enhance recognition performance under limited data conditions. While this strategy improves data efficiency, the model remains a deep transformer-based architecture with limited interpretability at the level of motion representation.

More broadly, existing approaches can be categorized into video-based and skeleton-based methods. Video-based approaches operate directly on RGB video streams and learn spatio-temporal representations using 3D convolutional networks or transformer architectures. For instance, Wei et al. (2023) employ Inflated 3D ConvNet (I3D) (Carreira & Zisserman, 2017) combined with a Multi-Stage Temporal Convolutional Network (MS-TCN) (Li et al., 2019). Their pipeline includes child localization followed by extraction of 3D spatio-temporal features and behavioral classification. Similarly, Ali et al. (2022) adopt a two-stream architecture (RGB and optical flow), integrating YOLOv5 (Jocher et al., 2020) and DeepSORT (Wojke et al., 2017) for child detection and 3D-CNN-based classification with late feature fusion. Explicit modeling of optical flow enhances sensitivity to motion dynamics. Yoo et al. (2024) follow a comparable strategy, combining a 3D-CNN backbone for spatio-temporal feature extraction with a Video Swin Transformer (Liu et al., 2022) for segment-level classification, thereby integrating convolutional motion encoding with transformer-based modeling of long-range temporal dependencies.

In contrast, skeleton-based approaches rely on joint coordinates and graph representations of human motion rather than raw RGB input. Ruan et al. (2023) propose a method based on ST-GCN (Yan et al., 2018) combined with contrastive learning, where behavior recognition is formulated as a sequence of binary decisions organized in a hierarchical tree. Instead of direct multi-class classification, the model identifies structural motion attributes—such as symmetry, periodicity, and dominance—which are subsequently aggregated to infer more complex behavioral patterns. This decomposition improves interpretability compared to purely end-to-end 3D-CNN or transformer-based models, as the intermediate nodes correspond to semantically meaningful motion characteristics.

Another important axis of differentiation concerns the nature of the video data: controlled diagnostic scenarios versus spontaneous, free behavior. In the works of Cai et al. (2022) and Prakash et al. (2025), behavioral analysis is conducted within structured diagnostic settings. In particular, Cai et al. (2022) propose a specifically designed recording protocol in which a parent uses a mobile phone or toy to guide the child's attention to predefined spatial positions (center, left, and right). From these controlled interactions, 709 features related to head pose, gaze direction, and facial expressions (extracted using OpenFace 2.0 (Baltrušaitis et al., 2018)) are computed. An HRC attention mechanism (Lan et al., 2019) is then applied to select the most discriminative features, followed by CNN-based binary classification. While such structured protocols reduce environmental variability and improve reproducibility, they limit generalization to unconstrained real-world settings.

Overall, current research can be structured along two main dimensions: the modeling paradigm (video-based vs. skeleton-based) and the data acquisition setting (controlled diagnostic interactions vs. spontaneous free behavior). The choice of approach reflects a trade-off between recognition accuracy, interpretability, data requirements, and practical deployability in real-world environments.

## 2 DATASET

In collaboration with correctional education specialists, twelve stereotypical motor behaviors observable in children under the age of three were identified. These behaviors reflect clinically relevant repetitive motor patterns and object-related actions frequently discussed in early ASD screening practice.

The selected behavioral categories are summarized in Table 1. The numbering corresponds to the internal annotation scheme used throughout the dataset.

To construct the dataset, 66 videos were recorded in medical centers. The recordings include 36 children. All videos were captured indoors in naturalistic play settings, where children interacted with toys or moved freely within the room.

Visual inspection of the raw material revealed several important limitations.

First, the recording conditions were not uniform. Some videos were captured using a static camera, while others involved camera motion. Camera displacement significantly reduces motion analysis

Table 1: Annotated stereotypical motor behaviors

| ID | DESCRIPTION |
|----|-------------|
| 1 | Hand flapping (one or both hands), possibly with an object |
| 2 | Whole-body rocking while standing or sitting |
| 3 | Repetitive locomotion along a trajectory (circular or linear) |
| 4 | Jumping in place or along a repetitive trajectory |
| 5 | Stepping from foot to foot repetitively |
| 6 | Walking on tiptoes |
| 7 | Unusual ("atypical") wrist and/or finger movements |
| 8 | Building a tower from toys or other objects |
| 9 | Arranging objects in a line (by shape or color) or repeatedly correcting alignment |
| 10 | Repetitive close inspection of objects (bringing them close to the face or moving the face around the object) |
| 11 | Repetitive manipulation of objects |
| 12 | Repetition of the same vocal sound |

reliability, since both pose and trajectory cease to be stable relative to the image plane. Rotations, shaking, or changes in distance introduce artificial motion patterns that distort trajectory estimation.

If the camera is positioned too far from the child, behavioral interpretation becomes unreliable. An example is shown in Figure 1, where the action cannot be unambiguously determined. The child may be manipulating objects, arranging them in a line, or building a tower, but the motion is visually indistinguishable.

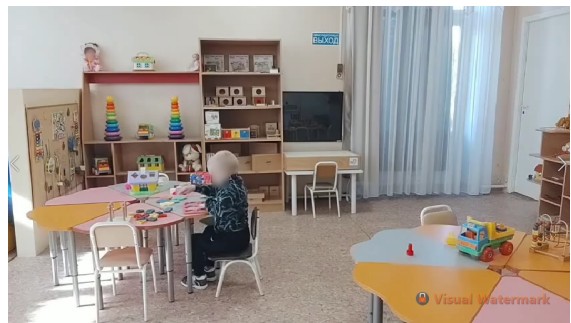

Figure 1: Camera positioned too far from the child.

Even with a fixed camera, occlusions occur frequently. The child may turn their back to the camera (Figure 2a), hide behind objects (Figure 2b), or be partially obstructed (Figure 2c).

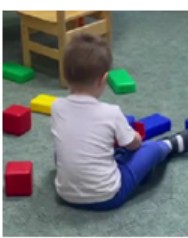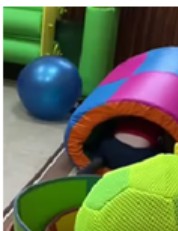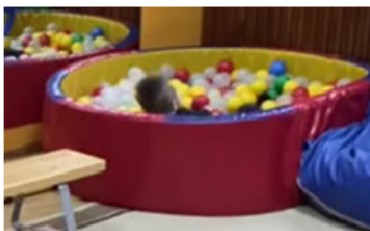

Figure 2: Occlusion scenarios: (a) back-facing pose, (b) hidden child, (c) object obstruction.

After applying automatic human pose estimation to extract skeletal keypoints, additional challenges emerged.

In cluttered scenes, a child surrounded by toys can be detected as a single merged entity (Figure 3a). Person identifiers (IDs) can change due to abrupt pose changes (Figure 3b). Additional individuals may appear in the frame and the child may not be fully visible (Figure 3c), complicating downstream tracking.

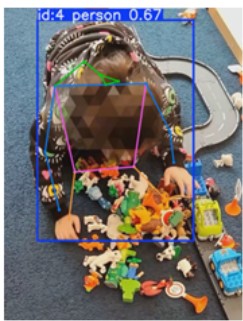 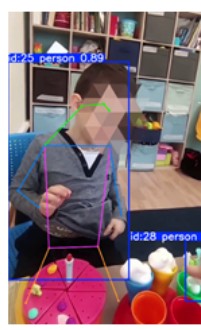 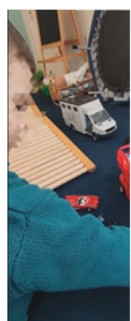

Figure 3: Detection issues: (a) merged detections, (b) ID switching, (c) additional person in frame.

Pose estimation algorithms may falsely detect human-like objects (e.g. dolls) as persons, typically with lower confidence scores (Figure 4a). However, realistic dolls may occasionally produce higher confidence values, increasing the ambiguity. Complex sitting or folded postures are also poorly reconstructed, particularly when the limbs are occluded.

Rarely overlapping skeletons are detected (Figure 4b), making it unclear which track should be continued. In one low-quality video (Figure 4c), automatic skeleton extraction failed. Since this video clearly demonstrated behavior ID 1 (hand flapping), six keypoints were manually annotated. Despite the short duration (3 seconds), manual correction required more than 30 minutes.

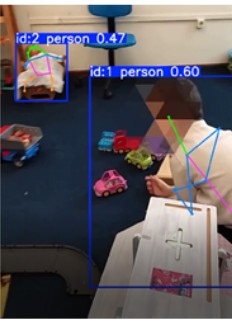 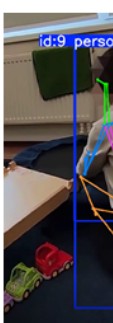 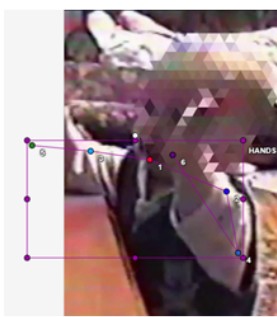

Figure 4: Complex recognition scenarios: (a) doll detected as person, (b) overlapping skeletons, (c) low-quality video.

When a child is far from the camera, confidence decreases (Fig. 5b). An additional unexpected limitation was the presence of mirrors. Reflections introduce artificial secondary skeletons (Fig. 5b), again affecting person identity consistency.

Figure 6(a) illustrates the number and total duration of examples for each behavioral category and the corresponding segment duration statistics (Fig. 6(b)). The statistics correspond to the original annotated segments; for model training, these segments were further divided into smaller samples according to a sliding-window size.

Despite the fact that 12 behavioral patterns were identified as characteristic of children with ASD, only two were considered in the present study. Some patterns cannot be reliably detected using skeletal keypoints alone. In particular, pattern 12 (repetitive vocalization) involves auditory information and therefore cannot be determined from video-based pose estimation. Patterns that involve significant spatial displacement of the child require accurate compensation for camera motion, which may become unreliable in cases of large camera movements. Behaviors involving interactions with external objects cannot be inferred from skeletal trajectories alone. Finally, patterns related to unusual

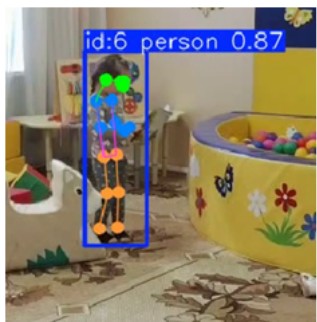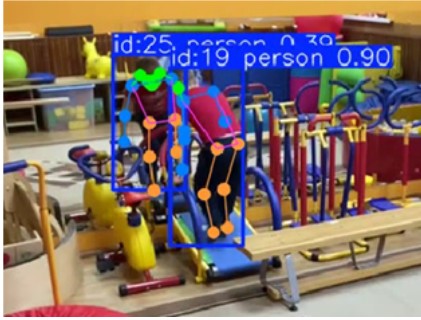

Figure 5: Recognition challenges due to distance (a) and mirror reflections (b).

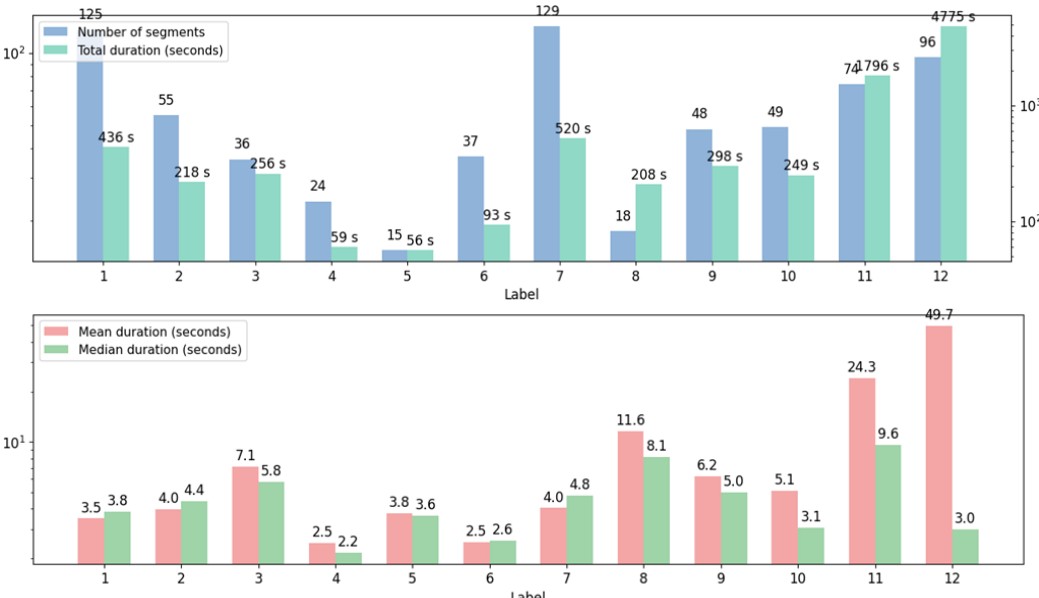

Figure 6: Number of examples for each behavioral category (a) and their duration (b).

finger movements require a more detailed hand keypoint model than the body pose representation used in this work. For these reasons, hand flapping and whole-body rocking were selected for analysis, as they are the most reliably detectable using skeletal motion dynamics.

# 3 METHODS

## 3.1 PROBLEM SETTING

We consider screening in unconstrained videos of spontaneous child behavior. Instead of analyzing the entire video at once, the system operates in a *sliding-window* regime: the input stream is decomposed into temporal segments (windows) of potentially different duration. Each window is processed independently to extract pose-based trajectories and predict the presence or absence of predefined motor stereotypies. This formulation naturally supports long recordings and allows to detect the most part of intermittent behaviors.

Let each window be associated with binary predictions for a set of target stereotypical patterns. If at least two distinct patterns are detected within the same window, the segment is labeled as high-risk. Thus, risk assessment is performed at the window level by aggregating multiple motor indicators while preserving temporal localization.

## 3.2 POSE ESTIMATION AND TRACKING

The first stage is pose estimation: for each frame we detect the child and estimate 2D keypoints. We employ YOLOv11-Pose, which predicts 17 COCO-format keypoints in the image plane. Although MediaPipe Pose provides a denser 33-keypoint representation, in our data YOLOv11-Pose was more robust to partial occlusions and boundary truncations, i.e., it maintained a more consistent subject localization when body parts left the frame. This qualitative difference is illustrated in Figure 7, where MediaPipe (a) may fail to localize the child correctly, while YOLOv11-Pose (b) remains stable under the same conditions. We additionally enable multi-frame tracking to obtain temporally consistent track IDs; among tested trackers, ByteTrack provided the most stable associations.

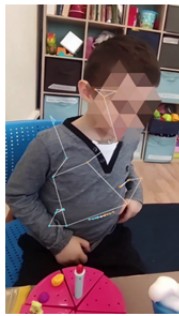 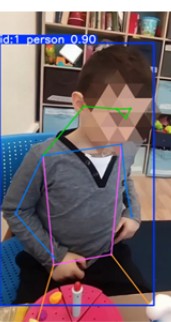

Figure 7: Pose estimation Yolov11Pose m bytetrack (a) vs MediaPipe Blazepose heavy (b).

## 3.3 KALMAN FILTERING OF KEYPOINT TRAJECTORIES

Raw keypoint trajectories exhibit frame-to-frame jitter due to measurement noise and occasional detector failures. In ASD-oriented motor stereotypy screening, such high-frequency noise may imitate low-amplitude repetitive motion, increasing false positives.

**State model.** For each coordinate ($x$ and $y$ processed independently), we use a constant-velocity state:

$$\hat{\mathbf{x}}_k = \begin{bmatrix} p_k \\ v_k \end{bmatrix}, \qquad \mathbf{P}_k = \begin{bmatrix} \Sigma_{pp} & \Sigma_{pv} \\ \Sigma_{vp} & \Sigma_{vv} \end{bmatrix}. \tag{1}$$

**Prediction.**

$$\mathbf{F}_k = \begin{bmatrix} 1 & \Delta t \\ 0 & 1 \end{bmatrix}, \qquad \hat{\mathbf{x}}_{k|k-1} = \mathbf{F}_k \hat{\mathbf{x}}_{k-1|k-1}, \tag{2}$$

$$\mathbf{P}_{k|k-1} = \mathbf{F}_k \mathbf{P}_{k-1|k-1} \mathbf{F}_k^\top + \mathbf{Q}_k. \tag{3}$$

**Correction.**

$$z_k = \mathbf{H}\mathbf{x}_k + v_k, \qquad \mathbf{H} = [1 \; 0], \qquad v_k \sim \mathcal{N}(0, \mathbf{R}_k), \tag{4}$$

$$\mathbf{K}_k = \mathbf{P}_{k|k-1} \mathbf{H}^\top \left( \mathbf{H}\mathbf{P}_{k|k-1}\mathbf{H}^\top + \mathbf{R}_k \right)^{-1}, \tag{5}$$

$$\hat{\mathbf{x}}_{k|k} = \hat{\mathbf{x}}_{k|k-1} + \mathbf{K}_k \left( z_k - \mathbf{H}\hat{\mathbf{x}}_{k|k-1} \right), \tag{6}$$

$$\mathbf{P}_{k|k} = \mathbf{P}_{k|k-1} - \mathbf{K}_k \mathbf{H}\mathbf{P}_{k|k-1}. \tag{7}$$

Therefore, we post-process each keypoint coordinate time series (Fig. 8a) with a Kalman filter. The effect of Kalman filtering is illustrated in Figure 8b.

## 3.4 METRIC NORMALIZATION BY INTER-SHOULDER DISTANCE

To reduce scale variability across videos and express coordinates in physical units, we normalize by the inter-shoulder distance.

$$d_{\mathrm{px}} = \sqrt{(x_{\mathrm{ls}} - x_{\mathrm{rs}})^2 + (y_{\mathrm{ls}} - y_{\mathrm{rs}})^2}, \tag{8}$$

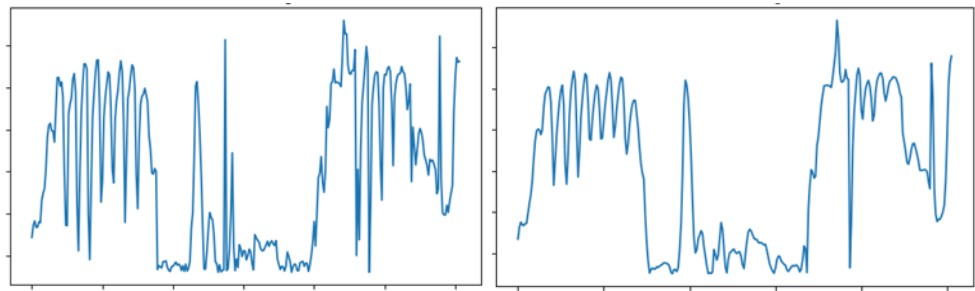

Figure 8: Elbow flexion angle: (a) before Kalman filtering; (b) after Kalman filtering.

and use $\bar{d}_{\mathrm{px}}$ averaged over frames where both shoulders exceed a confidence threshold (default $0.5$). For age $a$ (years), the reference distance is

$$d_{\mathrm{ref}}(a) = \begin{cases} 0.24, & a < 2, \\ 0.32, & a > 6, \\ 0.24 + (a - 2) \cdot \dfrac{0.08}{4}, & 2 \leq a \leq 6, \end{cases} \tag{9}$$

with the scale factor and conversion

$$s = \frac{d_{\mathrm{ref}}(a)}{\bar{d}_{\mathrm{px}}}, \qquad (x_{\mathrm{m}}, y_{\mathrm{m}}) = (s\, x_{\mathrm{px}},\, s\, y_{\mathrm{px}}). \tag{10}$$

### 3.5 FEATURE EXTRACTION

We study two motor stereotypy patterns: *(i)* hand flapping/shaking and *(ii)* whole-body rocking. For each windowed segment we compute physically interpretable features from wrist (hands) and torso trajectories.

All features listed in Tables 2 and 3 are computed from normalized 2D trajectories $\mathbf{p}(t) = (x(t), y(t))$ extracted within a sliding window segment of duration $T$ seconds containing $N$ samples.

Cycle-related features are derived from local extrema of the vertical coordinate $y(t)$. Here $P$ and $V$ denote the sets of local maxima and minima indices, respectively. Individual cycle durations are denoted $\mathrm{dur}_i$, and $T_{\mathrm{cyc}}$ represents the total time occupied by cyclic motion within the segment. The feature $C_{reg}$ evaluates how regularly events repeat over time.

Amplitude features describe spatial displacement. $\bar{A}_x$ and $\bar{A}_y$ denote mean absolute horizontal and vertical amplitudes, while $A_{\mathrm{max}}$ corresponds to the maximum peak-to-valley displacement. The quantity $\sigma_A$ measures variability of successive oscillation amplitudes.

Spectral characteristics are computed from the magnitude of the discrete Fourier transform $S(f)$ over the analyzed frequency range $F$. The subset $FR$ denotes the predefined repetition band (e.g., $2$–$5\,\mathrm{Hz}$). So $E_{FR}$ denotes ratio of power of the particular spectrum. Spectral entropy is computed from normalized spectral magnitudes.

Kinematic features are based on the velocity vector $\dot{\mathbf{p}}(t)$, while spatial descriptors rely on principal component analysis, where $v_1$ denotes the first eigenvector. The symbol $\rho$ represents the correlation between hand and torso vertical motion.

For whole-body rocking features (Table 3), $\mathbf{c}$ denotes the centroid of the torso trajectory, $r_i$ are distances from trajectory points to the centroid, $\sigma_r$ - the standard deviation of the radius, $\kappa$ - the value of the curvature along the trajectory. The term $d_{\mathrm{start-end}}$ refers to the distance between the first and last trajectory points, used to detect closed loops. For each pair of segments similarity measures are calculated in one-dimensional ($\mathrm{sim}_{1D}$) and two-dimensional ($\mathrm{sim}_{2D}$) spaces after applying principal component analysis (PCA). One–dimensional space is a segmented 1D projection of the trajectory (the main PCA component). The normalized similarity in two-dimensional space (2D) is calculated in a similar way for two-dimensional trajectories of the corresponding segments. $\mathrm{baf}$ is combined indicator of movement along a repeating trajectory ("back and forth") for the signal $\mathrm{a}(t)$ and the inverted $\mathrm{b}_{\mathrm{ref}}(t) = \mathrm{b}(-t)$

Similarity-based features are computed after temporal alignment of compared subsegments; RMSE denotes the root-mean-square error. The symbol $\tau^*$ indicates the first significant peak of the auto-correlation function, and $f_s$ denotes the sampling frequency.

Table 2: Hand (wrist) features with mathematical definitions

| CYCLES & AMPLITUDE | | FREQUENCY & SPECTRUM | |
|---|---|---|---|
| $N_{\text{cycles}}$ | $\min(|P|, |V|)$ | $f_{\text{dom}}$ | $\arg\max_{f \in F} |S(f)|$ |
| $f_{\text{mean}}$ | $\dfrac{N_{\text{cycles}}}{T}$ | $\sigma_f$ | $\sqrt{\dfrac{1}{M} \sum (f_i - \bar{f})^2}$ |
| $\bar{A}_y$ | $\dfrac{1}{N} \sum |y_i - \bar{y}|$ | $E_{FR}$ | $\dfrac{\sum_{f \in FR} |S(f)|^2}{\sum_f |S(f)|^2}$ |
| $A_{\max}$ | $\max |y(t_k^p) - y(t_k^v)|$ | $C_{reg}$ | $\max\left(0, \ 1 - \frac{\sigma_{\Delta t}}{\mu_{\Delta t}}\right)$ |
| $A_{\max} f_{\text{mean}}$ | product | $\text{fft}_{\text{peak}}$ | $\dfrac{\max_f |S(f)|}{\sum_f |S(f)|}$ |

| SPEED & ENERGY | | SPATIAL & CORRELATION | |
|---|---|---|---|
| $v_{\max}$ | $\max_t \|\dot{\mathbf{p}}(t)\|$ | $\theta_{\text{PCA}}$ | $\arccos(|v_1^{(y)}|) \dfrac{180°}{\pi}$ |
| $E_{\text{motion}}$ | $\sum_t \|\dot{\mathbf{p}}(t)\|^2$ | $\rho$ | $|\text{corr}(y_h, y_t)|$ |
| $R_{\text{vert}}$ | $\dfrac{\bar{A}_y}{\bar{A}_x + \bar{A}_y + \varepsilon}$ | $\sigma_A$ | $\sqrt{\dfrac{1}{N} \sum (A_i - \bar{A})^2}$ |

Table 3: Whole-body rocking features with mathematical definitions

| GENERAL CYCLICITY | | CIRCULAR MOTION | |
|---|---|---|---|
| $N_{\text{cycles}}$ | $f_{\text{dom}} T$ | $\bar{r}$ | $\dfrac{1}{N} \sum r_i$ |
| $R_{\text{cyc}}$ | $\dfrac{T_{\text{cyc}}}{T}$ | $\sigma_r$ | $\sqrt{\frac{1}{N} \sum (r_i - \bar{r})^2}$ |
| $\sigma_{\text{dur}}^2$ | $\text{Var}(\text{dur}_i)$ | $\kappa$ | $\dfrac{|x'(t)y''(t) - y'(t)x''(t)|}{\left(x'(t)^2 + y'(t)^2\right)^{3/2}}$ |
| $A_{\text{vert}}, A_\theta$ | $\max(y) - \min(y)(\max(\theta) - \min(\theta))$ | closed | $d_{\text{start-end}} < 0.3\bar{r}$ |

| REPEATED SEGMENTS | | PERIODICITY | |
|---|---|---|---|
| $\text{sim}_{1D}$ | $\max\left(0, 1 - 2\dfrac{\text{RMSE}}{\max(\text{range})}\right)$ | $f_{\text{dom}}$ | $\arg\max_{f \in F} |S(f)|$ |
| $\text{sim}_{2D}$ | $\max\left(0, 1 - 2\dfrac{\text{RMSE}(x, y)}{\frac{1}{2}(range_x + range_y)}\right)$ | $E_{FR}$ | $\dfrac{\sum_{f \in FR} |S(f)|^2}{\sum_f |S(f)|^2}$ |
| baf | $0.5 - \text{RMSE}(a, b_{\text{ref}}) + 0.5 \, \text{sim}_{corr}$ | $T_{\text{acorr}}$ | $\dfrac{\tau^*}{f_s}$ |
| $\text{fft}_{\text{peak}}$ | $\dfrac{\max_f |S(f)|}{\sum_f |S(f)|}$ | $N_{\text{acorr}}$ | $\dfrac{T}{T_{\text{acorr}}}$ |

## 4 EXPERIMENTS

The paper considers classical machine learning algorithms, many of which provide built-in feature importance estimation. Due to the limited dataset size, model evaluation is performed using grouped 5-fold cross-validation.

Several trajectory segments analyzed in this work originate from the same video, which may introduce correlations between training and test samples. To avoid optimistic bias, a grouped version of K-fold cross-validation is applied, where all segments from the same video belong to the same fold and each group appears in the test set exactly once.

We report standard binary classification metrics:

$$\text{Accuracy} = \frac{TP + TN}{TP + TN + FP + FN}, \qquad \text{Precision} = \frac{TP}{TP + FP}, \qquad \text{Recall} = \frac{TP}{TP + FN},$$

$$\text{F1} = 2 \cdot \frac{\text{Precision} \cdot \text{Recall}}{\text{Precision} + \text{Recall}} \tag{11}$$

The training pipeline is as follows:

1. The dataset is divided into 5 nonoverlapping folds;
2. Training and test sets are formed;
3. Features in the training set are normalized (except for tree-based models);
4. The same normalization parameters are applied to the test set;
5. The model is trained on the training data;
6. Predictions are obtained for the test set;
7. TP, TN, FP, and FN values are recorded together with SHAP values;
8. After all folds are processed, Accuracy, Precision, Recall, and F1-score are computed.

### 4.1 MODELS AND CONFIGURATION

We evaluate Logistic Regression (LR), Decision Tree (DT), Random Forest (RF), Gradient Boosting (GB), Support Vector Machine (SVM), and a small MLP (Table 4). For scale-sensitive models (LR, SVM, MLP), features are standardized using training-fold mean and variance and then applied to the test fold; tree-based models operate on raw feature scales.

Table 4: Model configuration used in experiments

| MODEL | CONFIGURATION |
|-------|---------------|
| LR | L2 regularization, $C = 1.0$, class_weight=balanced, solver=liblinear, max_iter=1000 |
| DT | Standard CART; pruning/limits tuned to avoid trivial overfitting |
| RF | Bootstrap sampling; random feature subsets; ensemble of decision trees |
| GB | Gradient boosting over shallow trees (small-depth regime) |
| SVM | Linear; standardized features; tested with full and selected feature sets |
| MLP | Hidden layers (32,16); ReLU; Adam; L2 weight decay $= 0.01$; lr $= 0.001$; epochs $= 200$ |

### 4.2 RESULTS

Hand flapping may be unilateral or bilateral, while labels are provided per segment without specifying the active hand. Concatenating left/right features can bias the classifier toward the dominant hand in the dataset and reduce sensitivity when only one hand moves. Training two separate hand-specific classifiers is not feasible without hand-specific labels.

We aggregate 16 hand features using element-wise max pooling:

$$\mathbf{f}_{\max} = \max\left(\mathbf{f}_L, \mathbf{f}_R\right) \quad \text{(element-wise).} \tag{12}$$

We additionally report SHAP-based feature importance directly within the performance tables in order to jointly present predictive quality and model interpretability (Table 5). For rocking, a single torso trajectory is used, hence no cross-limb aggregation is required. All 16 torso features are fed directly into the classifiers (Table 6).

Table 5: Hand flapping/shaking performance (grouped 5-fold CV; max-pooled features).

| MODEL | ACC | REC | F1 | TOP SHAP FEATURES |
|---|---|---|---|---|
| Logistic Regression | 0.921 | 0.905 | 0.927 | $f_{\text{dom}},\ \sigma_f,\ \frac{N_{cycles}}{T}, v,\ A_{\max}$ |
| Decision Tree | 0.921 | 0.857 | 0.923 | $f_{\text{dom}}$ |
| Random Forest | 0.921 | 0.857 | 0.923 | $f_{\text{dom}}$ |
| Gradient Boosting | 0.947 | 0.905 | 0.950 | $f_{\text{dom}}, C_{reg}, E_{FR},\ N_{\text{cycles}},\ f_{\text{mean}}$ |
| SVM (all features) | 0.895 | 0.905 | 0.905 | $C_{reg},\ f_{\text{dom}},\ A_x, \rho,\ \theta_{\text{PCA}}$ |
| MLP | 0.921 | 0.905 | 0.927 | $C_{reg}, f_{\text{dom}}, fft, v, A_{max}$ |

Table 6: Whole-body rocking performance (grouped 5-fold CV; torso features).

| MODEL | ACC | REC | F1 | TOP SHAP FEATURES |
|---|---|---|---|---|
| Logistic Regression | 0.885 | 0.762 | 0.821 | $A_{vert},\ N_{cycl},\ C_{reg},\ A_{\theta},\ T_{cyc}$ |
| Decision Tree | 0.754 | 0.714 | 0.667 | $N_{\text{cycles}},\ A_{vert},\ \text{fft}_{\text{peak}},\ T_{cyc},\ E_{\text{FR}}$ |
| Random Forest | 0.836 | 0.762 | 0.762 | $A_{vert},\ \text{fft}_{\text{peak}},\ N_{\text{cycles}},\ T_{cyc},\ R_{cyc}$ |
| Gradient Boosting | 0.918 | 0.810 | 0.872 | $\text{fft}_{\text{peak}},\ A_{vert},\ N_{\text{cycles}},\ T_{cyc},\ \bar{\kappa}$ |
| SVM | 0.803 | 0.667 | 0.700 | $A_{vert},\ \text{sim}_{2D},\ \text{fft}_{\text{peak}},\ N_{cyc},\ \bar{\kappa}$ |
| MLP | 0.902 | 0.810 | 0.850 | $A_{vert},\ \text{fft}_{\text{peak}},\ N_{circ},\ \bar{\kappa},\ \text{sim}_{2D}$ |

## 5 CONCLUSION

In this work, we constructed and analyzed a dataset of spontaneous child behavior comprising 66 annotated videos of 36 children. The dataset has been specifically designed for the study of Stereotypical Motor Movements (SMM) in unconstrained settings and will continue to be expanded to increase both behavioral diversity and statistical robustness.

We proposed a pose-based pipeline for window-level detection of ASD-related motor patterns using interpretable kinematic and spectral features. For two target behaviors—hand flapping and whole-body rocking—we obtained strong performance under grouped 5-fold cross-validation. In particular, gradient boosting achieved Accuracy 0.947 and F1 0.950 for hand flapping detection, and Accuracy 0.918 with F1 0.872 for whole-body rocking.

SHAP-based analysis identified the most influential features for SMM detection, with dominant frequency, cycle regularity, vertical amplitude, and spectral energy ratios emerging as consistently significant across models. These findings confirm the relevance of periodic and cyclic descriptors for modeling stereotypical motor behavior.

## 6 DISCUSSION

In future work, we plan to extend the current solution by incorporating the underexplored stereotypical behaviors and developing an adaptive boosting ensemble that combines heterogeneous modalities, including skeletal motion features, visual embeddings, and questionnaire data. Since different stereotypical behaviors contribute unequally to prediction quality, the ensemble will adaptively reweight modality-specific classifiers, emphasizing diagnostically informative patterns while reducing the influence of redundant cues. This modality aware reweighting strategy is expected to improve robustness, interpretability, and overall performance in early ASD risk screening.

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
