# OpenReview forum: "Interpretable Time-Series and Trajectory Analysis for Detection of Autism-Related Motor Behaviors"
_mathai.club/MathAI/2026/Conference — 2026 Oral_

### Official Review · Reviewer_UgVD · 2026-03-10
**The abstract “Interpretable Time-Series and Trajectory Analysis for Detection of Autism-Related Motor Behaviors” describes the detection of stereotypical movements in children with autism spectrum disorder (ASD) based on the analysis of kinematic and spectral characteristics of motion.**

**Rating:** 8
**Confidence:** 4

**Review:**

The abstract “Interpretable Time-Series and Trajectory Analysis for Detection of Autism-Related Motor Behaviors” describes the detection of stereotypical movements in children with autism spectrum disorder (ASD) based on the analysis of kinematic and spectral characteristics of motion. The data used in this study consist of videos of children recorded by experts in controlled playroom environments. The experiments were conducted on a dataset comprising 103,000 frames from 42 videos of 18 children. Using the YOLOv11Pose model, the child’s pose is extracted from each video frame in the form of skeletal keypoints. The contribution of individual features to the classifiers is analyzed using feature importance measures, such Pearson correlation. By classifying the segments of the hand trajectories (shaking/no shaking) using the threshold method, the metrics Accuracy: 0.763, Precision: 0.762, Recall: 0.800, F1: 0.780 were obtained. The use of machine learning algorithms has increased the F1 metric to values of ~ 0.8 – 0.84.
The Abstract is well -written and present valuable results for readers.

---

### Official Review · Reviewer_1QAc · 2026-03-11
**Interpretable Time-Series and Trajectory Analysis for Detection of Autism-Related Motor Behaviors: A promising approach, requiring further development**

**Rating:** 6
**Confidence:** 4

**Review:**

This paper addresses the important problem of early autism spectrum disorder (ASD) detection in children by analyzing stereotypical motor patterns. The authors propose an approach based on extracting skeletal keypoints from videos using YOLOv11Pose, computing interpretable kinematic and spectral features (frequency characteristics, cyclicity, trajectory shape), and subsequent classification of temporal windows with machine learning methods (Logistic Regression, Gradient Boosting). Experiments are conducted on a dataset of 42 videos from 18 children. The best results are achieved for hand flapping detection (F1 = 0.84) and whole-body rocking (F1 = 0.87).

Pros:
1.Early ASD diagnosis is a priority in modern medicine, and using video analysis for this task represents a promising direction.
2.The proposed approach employs physically meaningful features: dominant frequency, movement amplitude, trajectory radius, coefficient of variation. This is crucial for medical applications where explainability is required.
3.The authors clearly identify the key characteristics of stereotypies (cyclicity, periodicity, repetitiveness), which logically leads to the selection of appropriate features.
4.The paper is formatted according to the conference template.

Cons (require improvement):
1.Incomplete experimental results. The paper introduces 12 behavioral categories (Table 1, page 3), but the results section (Section 4) only provides metrics for two of them: hand flapping and whole-body rocking. Results for the remaining 10 behaviors are missing. If the authors focus only on two types of stereotypies, this should be clearly stated in the problem formulation.
2.Unclear validation protocol. The paper does not specify whether a subject/video-wise data split was used to prevent data leakage, where segments from the same video could appear in both training and test sets. Without this, the results might be optimistically biased, as the model could memorize specific children rather than learning generalizable patterns of stereotypies.
3.Lack of comparison with existing methods. The authors do not compare their approach with state-of-the-art methods, even though they mention several in the literature review (page 2): 3D-CNN, MS-TCN, ST-GCN, Video Swin Transformer. Without such comparison, it is impossible to assess whether the proposed approach outperforms or underperforms existing solutions. Even a comparison with a simple baseline (e.g., a 3D-CNN trained on frames) would significantly strengthen the paper.
4.Lack of dataset statistics. Class distribution, segment durations, and information about data imbalance are not provided. This information is critical for proper interpretation of the reported metrics.

---

### Official Review · Reviewer_FpKg · 2026-03-12
**Interpretable Analysis of Motor Stereotypies in ASD**

**Rating:** 7
**Confidence:** 4

**Review:**

This paper proposes a method for detecting stereotypic movements in autism based on the analysis of skeletal trajectories. The authors reduce the task to feature extraction using "interpretable kinematic and spectral descriptors capturing circularity, repetition, and periodicity". Risk assessment is conducted in a sliding window, where "risk assessment is performed at the window level by aggregating multiple motor indicators".
Strengths An undeniable advantage of the study is the transparency of decision-making due to the use of classical machine learning algorithms and the SHAP method. The authors apply a rigorous testing methodology, implementing "grouped 5-fold cross-validation" to prevent data leakage between frames. The proposed pipeline demonstrates excellent metrics, as for classifying hand flapping "gradient boosting achieved the best performance Accuracy 0.947, F1 0.950". Data preprocessing is executed well through Kalman filter smoothing and physical normalization of coordinates. Using the YOLOv11-Pose detector proved to be a successful decision, as it maintains stability during partial occlusions in natural conditions.
Weaknesses However, the reviewed work also has several notable drawbacks. Out of the twelve types of stereotypic behavior declared at the beginning, only two were experimentally validated. The system is vulnerable in cluttered scenes because "Pose estimation algorithms may falsely detect human-like objects e.g. dolls as persons". The presence of mirrors in playrooms breaks tracking, as "reflections introduce artificial secondary skeletons". Aggregating the features of the left and right hands obscures important details, preventing the model from distinguishing unilateral and bilateral movements. In cases of annotation failure, resource-intensive intervention is required, where "manual correction required more than 30 minutes" even for a short video clip. The dataset size is limited to a sample of 36 children, significantly reducing confidence in the algorithm's generalization ability. The paper also lacks a direct comparison of the proposed pipeline with baseline solutions on other open datasets.
Evaluation and Ratings The proposed interpretable approach has high clinical significance for early screening tasks. The study lays a solid foundation for developing lightweight diagnostic systems that do not require massive computational power. The paper is rated 7 (Good paper, accept), as the authors proposed a high-quality and technically sound method. The confidence score is 4 (The reviewer is confident but not absolutely certain), since the algorithm's testing is limited to a private set of videos.

---

### Decision · Program_Chairs · 2026-03-14

**Decision:**

Accept (Oral)

**Comment:**

Dear Author(s),

On behalf of the Program Committee of the International Conference on Mathematics of Artificial Intelligence (MathAI 2026), we are pleased to inform you that your paper has been accepted for an oral presentation at MathAI 2026.

Your paper was evaluated through a rigorous two-stage review process involving both automated screening and expert review by members of the Program Committee. The reviewers recognized the quality and contribution of your work.

Presentation details:

- Format: Oral presentation (15–20 minutes + 5 minutes Q&A)
- Mode: You may present either in person (offline) at the conference venue in Sirius, Russia, or remotely via Zoom. Please indicate your preferred mode when confirming your participation.
- Conference dates: Marh 30 - April 3, 2026
- Website: https://mathai.club

Next steps:

1. Please confirm your participation and presentation mode by replying to this email mathai.club@yandex.ru no later than March 15, 2026 18:00 Moscow time.
2. If you plan to attend in person, the organizing committee will provide accommodation details separately.
3. Please prepare your final camera-ready manuscript according to the formatting guidelines available at https://mathai.club and upload it to OpenReview by March 15, 2026 18:00 Moscow time.

Should you have any questions regarding the program, logistics, or your presentation slot, please do not hesitate to contact us.

We look forward to your contribution to MathAI 2026.

With kind regards,

MathAI 2026 Program Committee
International Conference on Mathematics of Artificial Intelligence
https://mathai.club
OpenReview: https://openreview.net/group?id=mathai.club/MathAI/2026/Conference
Telegram: https://t.me/MathAI_club
Email: mathai.club@yandex.ru